# Eating Disorders and Disturbed Eating Behaviors Underlying Body Weight Differences in Patients Affected by Endometriosis: Preliminary Results from an Italian Cross-Sectional Study

**DOI:** 10.3390/ijerph20031727

**Published:** 2023-01-18

**Authors:** Fabio Panariello, Gianluca Borgiani, Concetta Bronte, Giovanni Cassero, Giulia Montanari, Marcella Falcieri, Michele Angelo Rugo, Ornella Trunfio, Diana De Ronchi, Anna Rita Atti

**Affiliations:** 1Department of Biomedical and Neuromotor Sciences (DIBINEM), University of Bologna, 40125 Bologna, Italy; 2Endometriosis and Pelvic Pain Outpatient Clinic (Regional Diagnostic and Therapeutic Path), Family Care Center (Local Heath Authority of Bologna, Department of Primary Care), 40123 Bologna, Italy; 3Eating Disorder Clinic Residenza Gruber, 40141 Bologna, Italy

**Keywords:** endometriosis, BMI, pain, eating disorders, disturbed eating behaviors, emotional eating attitudes

## Abstract

This study aimed to characterize the prevalence of eating disorders (EDs), disturbed eating behaviors (DEBs), and emotional eating attitudes (EEAs) among patients affected by endometriosis in order to understand a potential crosslink between this impacting gynecological disease and a Body Mass Index shift. A total of 30 patients were recruited at an endometriosis outpatient clinic in Bologna and were assessed by using standardized instruments and specific questionnaires for EDs, DEBs, and EEAs. Sociodemographic information and endometriosis clinical features and history information were collected by adopting a specific questionnaire. Retrospective reports of lifetime Body Mass Index (BMI) changes, current BMI, peak pain severity during the last menstrual period, and the average of pain intensity during the last intermenstrual period were used for a correlation with the mean score from eating-behavior scales’ assessment. The preliminary results indicate that, although only 3.33% of endometriosis patients are affected by ED, statistically significant differences at the mean scores of DEBs and EEAs assessment scales were found by stratifying patients on the basis of BMI levels at risk for infertility and coronary heart disease and on the basis of moderate/severe pain levels. The enrichment of the sample size and the recruitment of the control group to complete the study enrollment will allow us to investigate more complex and strong correlation findings and to assess the prevalence of EDs among endometriosis patients.

## 1. Introduction

Endometriosis is a complex clinical syndrome that is characterized by an estrogen-dependent chronic inflammatory process that affects primarily pelvic tissues, including the ovaries, with impaired reproductive fitness and general health [1]. The classic definition of endometriosis includes the ectopic presence of endometrial glands and stroma [2]. The main symptoms are dysmenorrhea, dyspareunia, chronic pelvic pain, irregular uterine bleeding, and/or infertility. It is estimated that it affects approximately 7–10% of women with a clinically relevant condition in about 3% of fertile-age women [3]. The real prevalence of endometriosis is difficult to establish because of the possibility of several asymptomatic disease cases [4].

Endometriosis is considered a multifactorial disease with an unclear etiology that recognizes immune, endocrine, and genetic risk factors. In recent years, the theory that the cause of ectopic migration of endometrial tissues can be linked to abnormalities of the immune system and, in particular, to the lack of cellular immunity has become increasingly important [5]. Taking everything into consideration, it seems that the interplay of immune, endocrine, genetic, anatomical, and environmental factors plays a key role in the overall etiopathogenetic mechanism [6]. There has also been an increased interest in the identification of modifiable risk factors for endometriosis, including those related to nutrition and weight changes [7].

On one side, endometriosis’ symptoms (above all, pelvic and abdominal pain) may affect patients’ eating attitudes, often leading to food restriction [8] and hesitating in DEBs and/or EDs [9,10]. With regard to pain in particular, a growing body of evidence suggests that malnutrition, like restrictive behaviors in general, can have a significant impact on the onset and maintenance of chronic non-cancer pain. For this reason, the role of nutrition is gaining increasing attention in the management of clinical conditions characterized by chronic pain [11]. On the other hand, diet and eating habits can influence the risk of developing endometriosis [12,13,14].

Indeed, several lines of evidence also suggest that endometriosis’ patients commonly have a lower Body Mass Index (BMI) than their matched controls [15,16] or are underweight [17,18]. The interpretation of this evidence would be based on a double hypothesis: (a) retrograde menstruation could be facilitated in non-obese subjects due to reduced intra-abdominal pressure; and (b) the chronic pain that characterizes the typical symptomatology of endometriosis could lead to loss of appetite and restriction of food intake [19]. Furthermore, in a recent case-control study by Holdsworth-Carson et al., it is reported that women with endometriosis with a normal BMI amounted to 56% vs. 25.2% overweight, 14.3% obese, and 4.5% underweight [20].

By considering the role of immune system function in brain development, body weight, and appetite regulation, the disruption of the immune system has also been involved in causing and/or maintaining disturbed eating behaviors(DEBs) and eating disorders(EDs). Moreover, sex hormones play a role in eating attitudes: estrogen inhibits food intake, while progesterone and testosterone may increase appetite and reduce impulse control [21,22].

Furthermore, regarding the relationship between endometriosis and EDs, Gao et al. have found that women with a previous diagnosis of ED were more likely to be later diagnosed with endometriosis [9]. In addition, patients with endometriosis, especially those with pelvic pain, also have an increased vulnerability to various psychiatric disorders. There is, in particular, a tendency to develop an affective or anxiety disorder, and more generally, there is a tendency toward a dysregulation of the emotional-affective psychopathological dimension which can, in turn, influence the adoption of disturbed eating behaviors [23]. Disturbed eating behaviors (DEBs) are behavioral attitudes which consist in an abnormal eating pattern. Although patients who exert DEBs do not meet the diagnostic criteria for feeding and eating disorders according to DSM-5, it could predict the development of an ED clinical picture. It has therefore been highlighted how important it is to recognize them to prevent the development of a full-blown ED [24]. Dietary habits, sociocultural expectations, perceptions of body image, and psychological characteristics may lead to the development of DEBs [25,26].

Several mechanisms have been hypothesized to explain the complex relationship between maladaptive dietary attitudes and emotional distress [27]. Particularly, a number of findings suggest that one subtype of DEBs, the emotional eating behavior, may be an adaptive coping strategy in the case of external stressors, acting as a distractor or replacing very unpleasant emotions (anger, boredom, and loneliness) with more bearable ones [27,28]. Emotional overeating is also associated with persistent pain [29].

Eating disorders (EDs) are severe psychiatric disorders, often preceded by DEBs, characterized by abnormal nutrition that may or may not be associated with behaviors aimed at weight control. Obesity by itself is not classified as an eating disorder [30].

All eating disorders could involve significant impairment in physical health and psychosocial functioning [31].

Both the Diagnostics and Statistics Manual (DSM-5) and the International Classification of Diseases (ICD-11) comprehend six major eating disorders [32]. These include three main categorical diagnoses (anorexia nervosa, bulimia nervosa, and binge-eating disorder) and three disorders, previously considered primarily as childhood disorders (avoidant–restrictive food intake disorder, pica, and rumination disorder). The DSM-5 also provides subtype qualifiers, severity indicators, and remission definitions [31].

Anorexia nervosa (AN) is a severe mental disorder that is characterized by an intense fear of weight gain that is associated or not with a disturbed body image perception. It results in severe dietary restriction or severe weight loss due to compensatory behaviors (predominantly vomiting and/or excessive physical activity) that are associated with significantly disturbed cognitive and emotional functioning [33,34]. The common and serious medical complications that AN could determine affect all organs and systems, and they depend on malnutrition, weight loss, and purging behaviors [35].

Bulimia nervosa (BN) is characterized by recurrent binging episodes, which imply eating large amounts of food with loss of control, in mandatory combination with compensatory behaviors such as self-induced vomiting, extreme exercise, and inappropriate use of medications such as laxatives and/or diuretics. These behaviors result from a negative self-assessment of body weight, shape, or appearance [36].

Binge-eating disorder (BED) is characterized by the recurrence of binge-eating episodes related to distress and associated with lower frequency in compensatory behaviors compared to bulimia nervosa [37].

Both bulimia nervosa and binge-eating disorder might result in obesity (30–45%) and consequently in metabolism disturbances [38,39].

Eating disorders can strike at any age and may affect individuals of different genders, sexual orientations, and ethnicities. Adolescents, young adults, and females are particularly at risk, and the age of onset is generally lower for anorexia nervosa compared to bulimia nervosa or binge-eating disorder [40,41].

Udo and Grilo estimated lifetime and 12-month prevalence of anorexia nervosa, bulimia nervosa, and BED as follows: 0.80 and 0.05%, 0.28 and 0.14%, and 0.85 and 0.44%, respectively [42]. The prevalence of eating disorders has increased more than 25% in the past 10 years, but only about 20% of affected individuals actively seek help [43,44].

It is relevant to penetrate more deeply the knowledge about the prevalence of disturbed eating patterns because EDs and the aptitude for disturbed eating, such as DEBs, are associated with multiple physical complications that strongly affect physical health, as has been shown, for instance, in both males and females with type 1 diabetes [45].

Taking in account the complex interplaying between BMI and endometriosis on one side and the biological factors that the broad spectrum of disturbed eating behaviors and endometriosis share on another side, the aims of this study were (a) to identify the prevalence of EDs in endometriosis patients; and (b) to assess DEBs occurrence among endometriosis patients and to estimate the emotional eating behavior related to endometriosis’ symptoms that may, in turn, contribute to symptoms severity even worsening medical comorbidities. To achieve these aims, a cohort of patients affected by endometriosis was recruited at the Local Health Authority of Bologna, Italy.

Considering the importance of detecting DEBs as potential predictors of the risk of developing full-blown ED and the impact of nutrition on chronic pain, both aims of this study aspired to gain evidence that could be useful to inform public-health policymakers to implement ED and DEBs prevention policies with consequent health promotion. For example, a screening of altered eating behaviors can be hypothesized for all patients with endometriosis, as well as an integrated multidisciplinary approach that provides psychological support to help patients adopt more adaptive coping strategies and nutritional support to improve eating habits. It may also be helpful to pay more attention to controlling endometriosis-related symptoms such as pain.

## 2. Materials and Methods

### 2.1. Participants

The study population consisted of 30 females recruited at a specialist outpatient clinic for endometriosis treatment of the Local Health Authority of Bologna, Italy, between November 2021 and September 2022.The limited sample size is due to the restrictions imposed as a result of the coronavirus pandemic, which resulted in the temporary interruption of recruitment.

The inclusion criteria were patients aged between 18 and 50 years, fully fluent in Italian and English, with a documented diagnosis of endometriosis, regardless of the date of onset of symptoms and of the type of localization of the ectopic endometrial glands and stroma, and with or without a surgery related to endometriosis.

Subjects with updated diagnoses to meet DSM-5 criteria since 2013 for major psychiatric disorders (attention disorders (attention deficit-hyperactivity disorder, attentive, or combined types); mood depressive disorder; bipolar disorder of type I or type II; obsessive–compulsive disorder (OCD); and schizophrenia spectrum disorders (schizophrenia, schizoaffective, or delusional disorder, as well as organic brain disorders)) were excluded from the study.

The cohort was assessed by using a questionnaire regarding information on endometriosis history, treatment, and symptoms and on demographic and anthropometric details.

In addition, all recruited patients were assessed by using the26-item Eating Attitudes Test (EAT-26) [46,47], a short version of the original 40-item instrument [48], the Eating Disorder Examination Questionnaire (EDE-Q) [49], and the Binge Eating Scale (BES) to detect patients at risk for ED.

In order to measure disordered eating attitudes, defined as abnormal beliefs, thoughts, feelings, behaviors, and relationships regarding food, and to evaluate emotional eating, marked out as the frequency with which individuals have eaten, over the prior 28 days to the assessment, an unusually large amount of food given the circumstances in response to feelings of anxiety, sadness, loneliness, tiredness, anger, happiness, boredom, guilt, and physical pain [50], the Disordered Eating Attitude Scale (DEAS) and the Yale Emotional Overeating Questionnaire (YEOQ)were filled out by the recruited patients.

As the following flowchart (Figure 1) describes, the complete assessment of all rating scales was conducted in two different rounds, on 2 different days, separated by 1 week, due to the adaptability of the protocol to the limitation of outpatient clinic facilities and the time required to complete all rating scales and clinical interviews.

The study protocol, which was carried out according to the ethical standards of the 2013 Declaration of Helsinki, was approved by the Research Ethics Committee for the Local Health Authority of Bologna, Italy (N. 726-2021-OSS-AUSLBO-21109-ID 2855). All participants provided written informed consent before entering the study.

### 2.2. Measures

Residential students in psychiatry (G.B. and C.B.), trained by a senior psychiatrist (F.P.),assessed the recruited subjects via an interview and supported them in completing the following self-administered questionnaires: EAT-26, EDE-Q, BES, DEAS, and YEOQ.

EAT-26 [46] is a 26-item self-reported questionnaire that has been widely used to measure the symptoms and behaviors associated with EDs in both clinical and non-clinical settings. It includes three subscales: “Diet” (13 items), “Bulimia and Food Preoccupation” (6 items), and “Oral Control” (7 items).

Participants were asked to answer to the items by using a 6-point Likert scale with different choices, namely “Never”, “Rarely”, “Sometimes”, “Often”, “Very Often”, and “Always”. A score of cutoff(20) or higher indicates the presence of symptoms associated with eating problems that require attention and further investigation [48,51]. For the present study, the validated Italian version was used [52].

The EDE-Q [53] was used to evaluate the basic psychopathology of eating disorders. It is a self-assessment questionnaire consisting of 28 items that provides four subscale scores (Restraint, Eating Concern, Shape Concern, and Weight Concern) and the overall score, which is, in turn, the average score of the four subscales.

The DEAS, in agreement with Alvarenga, Scagliusi, and Filippi (2010), who developed this self-administered scale, was used for the assessment of dietary attitudes as a form involving beliefs, thoughts, feelings, behaviors, and relationship toward food [54]. This scale has been used for evaluating clinical and non-clinical populations, as many people experience distorted eating practices, beliefs, and feelings about food. The original scale has been psychometrically assessed and considered cohesive and valid. The DEAS contains 25 questions designed to be rated on a Likert-type scale, and the expected overall score can range from 37 to 190. The scale was initially validated among a sample of Brazilian female students, and the internal consistency found was 0.75. The DEAS includes five subscales: (1) relationship with food, (2) concerns about eating and body weight gain, (3) restrictive and compensatory practices, (4) feelings toward eating, and (5) idea of a normal eating [54]. Higher scores were suggestive of more negative and/or disordered eating attitudes.

With the aim of measuring the frequency of emotional overeating in response to different emotional states, the YEOQ, a 9-item self-report questionnaire, was used. This scale was derived from the Emotional Overeating Questionnaire, or EOQ, initially developed by Masheb and Grilo (2006) to assess overeating as a consequence of six emotional conditions (anxiety, sadness, loneliness, tiredness, anger, and happiness) [55]. Since its initial development, it has been rearranged, adding several elements to the original version, up to the drafting of the YEOQ, which, in addition to the initial 6 emotions, includes 3 additional items, which are boredom, guilt, and the physical pain. For each emotion, participants are asked the following: “On how many days out of the past 28 days have you eaten an unusually large amount of food, given the circumstances, in response to feelings of...” (one of the 9 emotions taken into consideration). The frequency of emotional overeating behavior is rated on a 7-point scale: 0 (no day), 1 (1−5 days), 2 (6−12 days), 3 (13−15 days), 4 (16−22 days), 5 (23−27 days), or 6 (every day).The YEOQ has good validity with measures of eating disorder symptomatology, including binge eating and eating concern, as assessed by the Eating Disorders Examination Questionnaire [56]. The YEOQ, in previous studies [57], showed good internal consistency with Cronbach’s alpha = 0.946.

The BES is a 16-item questionnaire that describes the behavioral manifestations (8 items), feelings, and the cognitive aspect (8 items) that coexist with binging episodes [58]. Each item has three or four weighted statements, and test subjects are asked to choose one. The total score of the BES is obtained by adding the values for the 16 items, and the range of scores varies from 0 to 46 [58,59]. The Italian BES version in obese patients was validated by Di Bernardo et al. [60]. According to the reference literature, the total BES score was considered to obtain a continuous measure of binge-eating trends. A BES score ≥ 17 is an indication of binge-eating symptoms, although there is no evidence showing that it can validly be adopted to diagnose binge-eating disorder according to the DSM-5.

In order to assess the anthropometric characteristics, BMI was calculated as weight (self-reported) in kilograms over height (self-reported) in meters squared (kg/m^2^).

Based on the BMI, the subjects recruited, according to the classification of the World Health Organization, were divided into the categories underweight (≤18.5 kg/m^2^), normal weight (18.5−24.9 kg/m^2^), overweight (25−29.9 kg/m^2^), and obese (≥30 kg/m^2^).

Consistent with the epidemiological evidence that reported a higher risk for coronary heart disease [61] and ovulatory infertility [62], starting at the upper range of normal, the BMI range for the whole sample was also divided into two additional classes, lower and higher than 22.4 kg/m^2^,to evaluate a similar effect of endometriosis on weight gain and body-weight-related health risk.

To assess the pain associated with endometriosis, the Numeric Rating Scale (NRS) was used. Patients were asked to indicate the peak pain intensity during the last menstrual period and the pain intensity average in the time interval between the last 2 menstrual periods. The use of NRS is recommended by the American Society for Reproductive Medicine (ASRM) for conducting studies in endometriosis [63]. It is an 11-point numerical rating scale, where 0 means “no pain” and 10 means “unbearable pain”. Based on the pain intensity, the sample was then divided into 4 subgroups identifying painfulness (NRS = 0), mild pain (1 ≤ NRS ≤ 5), moderate pain (6 ≤ NRS ≤ 7), and severe pain (NRS ≥ 8).These cutoffs are in agreement with Boonstra et al. [64], who pointed out that an NRS score ≥ 5 may contribute to pain-related malfunctioning in patients with high catastrophizing tendency, which, in turn, significantly mediated the relationship between persistent pain and emotional eating behavior according to Janke et al. [29].

### 2.3. Statistical Analysis

Continuous and categorical variables were described as means with standard deviations (SDs) and percentages, respectively. The Shapiro–Wilk test was performed to assess the normality distribution for continuous variables.

Comparative analyses of qualitative variables were performed with Pearson’s Chi-square test or Fisher’s exact test, and comparative analyses of quantitative variables were run with Student’s *t*-test. For comparative purposes, two different grouping strategies were adopted according to the BMI and NRS score.

As formerly reported, the participants were at first divided in four groups on the basis of the BMI based on WHO classification, and subsequently in two groups according to Shah et al., by using BMI = 22.4 kg/m^2^ as cutoff [61,62]. In both grouping strategies, the distribution’s rate of the sample in the four and two subcategories, respectively, was calculated. To assess how significant the differences between group means were in the sample stratification into two groups according to Shah, the Student’s *t*-test was performed. On the report of the NRS score, four subgroups were initially identified as previously described, and afterward the whole sample was divided into two categories identified by the NRS score cutoff of 6 according to Boonstra at al. [64], and the Student’s *t*-test was applied to determine whether these two groups expressed a significant difference between population means. Based on an a priori sample size calculation, it was not possible to perform a logistic regression for the minimum statistical power level. A bivariate Pearson correlation was conducted to assess the strength and direction of the linear relationship between NRS mean score during the time interval between the last 2 menstrual periods and YEOQ total score. A *p*-value of <0.05 was considered statistically significant.

All data analyses were conducted using SPSS version 28.0 software (SPSS Inc., Chicago, IL, USA).

## 3. Preliminary Results

From November 2021 to September 2022, a total of 30 endometriosis patients were recruited for the study; these are the patients recruited so far, for the restrictions imposed as a result of the coronavirus pandemic resulted in difficulties regarding outpatient clinic access and the consequent temporary interruption of recruitment. Sociodemographic and endometriosis-related clinical data (duration of illness, symptoms (since pain in the lower abdomen is one of the main symptoms), type of treatment, and use of anti-inflammatory medications) were collected, and the compilation of the battery of self-administered scales was obtained(EAT-26, BES, EDE-Q, DEAS, and YEOQ).

As shown in Table 1, the average age and average current BMI were 37.53 ± 9.04 and 23.25 ± 6.5, respectively. In total, 80% of participants were at healthy weight (BMI 18.5–25 kg/m^2^), while 16.67% were overweight or obese (13.33% and 3.33%, respectively) (Table 2).

By adopting the grouping strategy according to Shah et al., in order to better identify patients with a greater risk for coronary heart disease [61] and ovulatory infertility [62], 40% of our sample belonged to a higher risk category (Table 3).

Moreover, the average weight gain since the onset of the disease was 5.65 ± 5.85.

The mean duration of illness was 7.83 ± 5.89, but the average of the symptomatology onset was longer (19.03 ± 11.34). A total of 83.33% of the sample was on hormonal treatment, and the average treatment length was consistent with the illness’ duration (7.68 ± 7.28).

The mean NRS score of the last menstrual period and during the time interval between the last 2 menstrual periods was 8.13 ± 2.21 and 4.43 ± 2.71, correspondingly (Table 1).

Among the 30 included patients, only 1 (3.33%) gained a total score for both EDE-Q and BES, suggesting a presumed ED diagnosis of binge-eating disorder (BED).

To assess whether the risk of disturbed eating attitudes varied on the basis of body weight, all participants were divided into two combined weight status categories according to the BMI cutoff for coronary heart disease risk [61] and ovulatory infertility [62].

As shown in Table 4, the scores on the total BES scale (*p* = 0.018) and subscales (behavior (*p* = 0.025); feelings and cognition (*p* = 0.013)); the EAT-26Bulimia and Food Preoccupation subscale (*p* = 0.029); and the YEOQ items for sadness (*p* = 0.003), loneliness (*p* = 0.004), and physical pain (*p* = 0.005); the total scores obtained at the EDE-Q (*p* = 0.018) and at DEAS Questionnaire (total (*p* = 0.003) and Food Concern subscale scores (*p* = 0.024)] differed significantly between the two groups.

Since the pain is the most common symptom in endometriosis and it may explain disturbed eating attitudes and weight gain, our sample was divided into two subgroups, according to NRS score (≥6) during the time interval between the last 2 menstrual periods, in order to point out if two subsamples differed in the assessment of disturbed eating attitudes.

As shown in Table 5, two subgroups showed significant differences, such as BMI-based groups, in the BES Total score (*p* = 0.05) and Feelings and Cognition subscale (*p* = 0.010); EAT-26Bulimia and Food Preoccupation subscale (*p* = 0.029); YEOQ items for sadness (*p* = 0.037), loneliness (*p* < 0.001), and physical pain (*p* = 0.030); and in the total scores obtained for DEAS Questionnaire (*p* = 0.020). In addition, also significant was the mean score for other feelings related to emotional overeating evaluated on the YEOQ scale, such as anxiety (*p* = 0.001), anger (*p* = 0.003), boredom (*p* = 0.003), and guilt (*p* = 0.011), that may also account for a significant difference in the total YEOQ score (*p* < 0.001).

The Chi-square test showed no significant differences (*p* = 0.57) in the distribution of overweight/obese patients across groups with or without intensity of pain that significantly interferes with daily functioning (NRS cutoff ≥ 6).

A moderate positive correlation was found between the mean score at the NRS scale during the time interval between the last 2 menstrual periods and YEOQ total score (Pearson’s correlation coefficient (r) = 0.51; *p* = 0.01), as shown in the scatterplot carried out by using SPSS (Figure 2).

## 4. Discussion

The first aim of the study was to identify the EDs prevalence rate in the recruited endometriosis cohort patients. The results, at the moment, suggest that only 3.33% of the patients were affected by ED (BED), but the small sample size does not allow us to infer about the clinical significance of this result. The topic of comorbidity between EDs and endometriosis deserves to be explored since, evoking Gao et al., women with previous eating disorders were more likely to be subsequently diagnosed with endometriosis [9].

The secondary aim of the study was to find if disturbed eating behaviors may contribute to symptoms’ severity and worsen medical comorbidities.

The preliminary data from our study suggest an association between BMI at risk for coronary heart disease and ovulatory infertility, DEBs, and endometriosis. In total, 40% of the examined sample had a BMI greater than 22.4 kg/m^2^, which, according to Willett et al. [61], defines a higher potential risk for coronary heart disease and, on the report of Rich-Edwards et al. [62], increases the risk for ovulatory infertility.

The 41.7% of this “higher-risk” population had a BMI which indicates overweight or obesity condition according to WHO cutoffs (BMI ≥ 25 kg/m^2^).

Only in one case (3.33%)could the high-risk weight condition be considered the consequence of a BED, while in all other cases, our data suggest a correlation with DEBs.

The subpopulation with a BMI greater than 22.4 kg/m2 reported significantly higher mean scores for the total BES and Behavior and Feelings/Cognitive subscales, the Bulimia and Food Concern subscale on the EAT-26 questionnaire, and the total EDE-Q compared to the endometriosis population with a lower BMI. 

Furthermore, this portion of the sample was also characterized by significantly higher mean scores on the DEAS and on the YEOQ items for sadness, loneliness, and physical pain.

The different distribution of the DEAS mean suggests that patients with a BMI > 22.4 kg/m^2^ experienced distorted eating habits as a consequence of an altered cognitive construct involving beliefs, thoughts, feelings, behaviors, and relationship toward food. The significant differences reported at the scores of different items/feelings of the YEOQ questionnaire (sadness, loneliness, and physical pain) indicate, however, a role of emotional eating in weight gain according with previous evidence [50,65].

The prevalence rates of sadness and loneliness feelings have previously been reported to be significantly higher in patients with endometriosis who are also affected by rates of depression ranging from 38% to 86%, depending on chronic pelvic pain as the predominant symptom [66].

It is also interesting to note that recent findings suggest that depressive and anxiety symptoms in patients with endometriosis appear to be mediated by body image, self-criticism, and pain intensity, which, in turn, may influence eating attitudes [67].

Although our study recognizes the diagnosis of major depression as an exclusion criterion, subthreshold depressive symptoms may affect the patients recruited into the sample study and may justify altered eating habits in terms of emotional overeating. Furthermore, it is recognized that subjects with dysfunctional affective traits tend to adopt frequent eating-compensatory behaviors [68].

Several lines of evidence suggest that chronic pain may be a barrier to weight loss and that emotional eating may at least partially clarify the overlap between pain and excess weight [69,70,71,72,73,74,75,76]. How chronic pain may influence eating behavior and weight gain is still poorly understood. Primarily, eating and chronic pain are related, as both interact with motivational states that influence decision-making [77,78,79,80,81]. According to the “fear avoidance model”, pain may initially cause obesity through movement aversion as an adaptive behavior to avoid pain exacerbation, and subsequently, pain patients become intolerant of physical activity by mechanism of deconditioning [82]. An alternative explanation is that the brain plays a key role in regulating energy intake and expenditure [83] in patients with pain through the involvement of the limbic reward circuits [84,85]. Recent studies suggest that both chronic pain and obesity are characterized by a hypodopaminergic state in the meso-corticolimbic system [86,87,88] and impaired opioid transmission within the limbic system [89] and by anhedonia [90].

Our preliminary results suggest that, also in our sample, there is a statistically significant difference in physical pain YEOQ item mean score between endometriosis patients with higher BMIs compared to lower BMIs.

These findings may be considered extremely relevant by considering the clinical picture that characterizes patients affected by endometriosis since the pain is one of the main symptoms [91]. Indeed, dysmenorrhea is often the first symptom of the onset of endometriosis, and other pain symptoms may also occur over the time, such as non-menstrual pelvic pain, deep dyspareunia, dyschezia, and chronic pelvic pain [92]. In addition, pain can occur intermittently or continuously during the menstrual cycle in endometriosis patients [93].

Taking into consideration the relevance of the pain in endometriosis, in our study, we evaluated the differences in terms of DEBs between patients who presented a level of pain that significantly affected the daily functioning in high catastrophizing population and those with a lower level of pain (NRS ≥ 6) according to Boonstra et al. [64]. The 46.6% of the sample had an NRS score that affects daily functioning.

The subgroup with an NRS score that interfers with daily functioning showed significantly higher scores evaluated on the total BES and Emotions and Cognitive BES subscale, on the EAT-26 Bulimia and Food Concern subscale, and on the total score of the DEAS questionnaire.

Furthermore, in this cohort of patients, the total scores for almost all the feelings evaluated at the YEOQ scale (sadness, loneliness, physical pain, anxiety, anger, boredom, and guilt) were significantly higher, resulting in a significant difference in the total score. These findings are consistent with those of Masheb et al., who found that all YEOQ items were statistically and significantly correlated with the physical pain [76]. Furthermore, in our entire sample, NRS scores correlated statistically significantly with total scores at YEOQ.

The pain is one of the most common stressors associated with impaired eating behaviors. According to Janke et al., anxiety sensitivity and catastrophizing may significantly mediate the correlation between persistent pain and emotional eating behavior [29]. It is interesting to report that high levels of stress, in turn, are implicated in the development and exacerbation of pain and can modify the motility and permeability of the gastrointestinal tract, thus influencing gut microbiota deviations [94].

The gut microbiota is involved in a complex bidirectional communication system along the gut–brain axis and may contribute to the regulation of emotional behavior and cognition [95].

Deviations of the intestinal microbiota have also been related to endometriosis [96]. A possible speculative hypothesis could involve the role of intestinal microbiota deviations in the reciprocal correlation between endometriosis, pain, and the dysregulation of emotionally based eating behavior and EDs [95].

This study has several limitations. The first concerns the small sample size and the lack of the control group which limited both the possibility to extend the statistical analysis to more sophisticated correlation modeling methods to obtain predictors of weight gain in patients with endometriosis and the extensiveness of the results. The measurement of height and weight was introduced on the basis of the participants’ declarations and was not carried out during the consultation due to logistical and organizational difficulties. This represents a limitation of the study, and we have planned the organization of the recruitment prosecution in order to guarantee the detection of the parameters during the consultation.

The small sample size is partly due to the reduced patient’s accessibility due to the SARS-CoV-2-pandemic-related restrictions. The control group, on the other hand, is in the recruitment phase, so the goal is to extend the sample size with ongoing sampling and to perform the analysis of the results within a case-control design study. Other issues can be traced to the predictive value of emotional eating questionnaires. Indeed, some studies have failed to find increased eating among self-reported emotional eaters during times of stress or other negative moods [97].

However, it is possible that other mechanisms in addition to emotions may be associated with self-reported emotional eating and weight gain [97]. Further investigations will be needed to understand the association between emotional eating and emotional state.

The strengths of our study are the inclusion of patients with endometriosis without major psychiatric comorbidity and the representativeness of the included patients with regard to age, date of symptom onset, and type of endometriosis.

## 5. Conclusions

The results, albeit preliminary, suggest that patients with endometriosis tend to have BMI values that constitute a risk factor for coronary heart disease and ovulatory infertility. DEBs and, in particular, pain-related emotional overeating, may mediate the relationship between higher BMIs and endometriosis.

It sounds reasonable to hypothesize the opportunity to implement the offer of psycho-nutritional support in a multidisciplinary setting in endometriosis pathway of care in order to promote a healthier eating style, which, in turn, could lead to the promotion of public health through the reduction of the consequences of DEBs.

## Figures and Tables

**Figure 1 ijerph-20-01727-f001:**
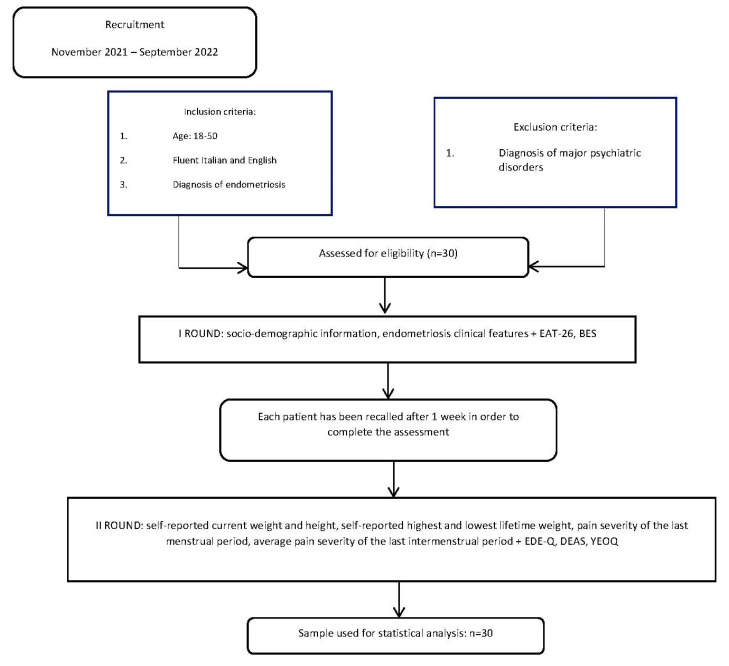
Flowchart of participant recruitment.

**Figure 2 ijerph-20-01727-f002:**
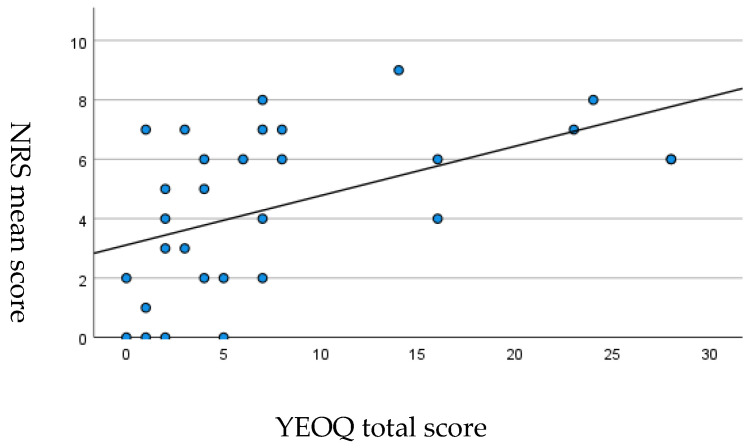
Correlation between NRS mean score during the time interval between the last 2 menstrual periods and YEOQ total score. Pearson’s correlation coefficient (r) = 0.51; *p* = 0.01.

**Table 1 ijerph-20-01727-t001:** Characteristics of the sample.

Parameter	Mean ± SD
Sample size (N)	30
Age	37.53 ± 9.04
Current BMI	23.25 ± 6.5
Weight gain	5.65 ± 5.85
Illness duration	7.83 ± 5.89
Symptoms duration	19.03 ± 11.34
Hormonal treatment duration	7.68 ± 7.28
Max pain during last menstrual period (NRS)	8.13 ± 2.21
Mean pain during last fertile month (NRS)	4.43 ± 2.71

**Table 2 ijerph-20-01727-t002:** Distribution of the sample in four groups based on BMI according to WHO categories.

Group	BMI Category	N (%)
1	Underweight (≤18.5 kg/m^2^)	1 (3.3%)
2	Normal weight (18.51−24.9 kg/m^2^)	24 (80%)
3	Overweight (25−29.9 kg/m^2^)	4 (13.3%)
4	Obesity (≥30 kg/m^2^)	1 (3.3%)

**Table 3 ijerph-20-01727-t003:** Distribution of the sample in two groups based on the BMI according to Shah cutoff.

Group	BMI Category	N (%)
1	BMI ≤ 22.4 kg/m^2^	18 (60%)
2	BMI > 22.4 kg/m^2^	12 (40%)

**Table 4 ijerph-20-01727-t004:** Mean scores differences between groups on BMI cutoff.

Scale	Subscale	BMI ≤ 22.4 (M ± SD)	BMI > 22.4 (M ± SD)	*p*-Value
	Current BMI	20.23 ± 1.36	27.27 ± 8.73	0.028 *
BES	Tot	3.17 ± 2.15	6.67 ± 9.44	0.018 *
	Behavior	2.28 ± 1.90	3.92 ± 5.47	0.025 *
	Feelings and Cognition	0.78 ± 0.88	3.00 ± 4.49	0.013 *
EAT-26	Tot	4.11 ± 2.68	6.25 ± 4.35	0.183
	Dieting	1.06 ± 1.06	3.92 ± 2.78	0.087
	Bulimia and Food Preoccupation	0.11 ± 0.471	0.67 ± 1.72	0.029 *
	Oral Control	2.67 ± 2.50	2.08 ± 1.78	0.099
YEOQ	Tot	6.89 ± 6.57	9.50 ± 10.47	0.061
	YEOQ-Anxiety	1.17 ± 1.51	1.33 ± 1.61	0.864
	YEOQ-Sadness	0.72 ± 0.075	1.25 ± 1.60	0.003 *
	YEOQ-Loneliness	0.56 ± 0.71	1.08 ± 1.56	0.004 *
	YEOQ-Tiredness	0.89 ± 1.49	1.17 ± 1.34	0.928
	YEOQ-Anger	0.78 ± 1.06	0.58 ± 1.00	0.691
	YEOQ-Happiness	0.83 ± 0.99	1.00 ± 0.85	0.733
	YEOQ-Boredom	0.89 ± 0.90	0.92 ± 1.31	0.400
	YEOQ-Guilt	0.44 ± 0.78	0.67 ± 0.99	0.453
	YEOQ-Physical Pain	0.61 ± 1.20	1.50 ± 1.98	0.005 *
EDE-Q	Tot	1.16 ± 0.715	1.83 ± 1.31	0.018 *
DEAS	Tot	65.89 ± 10.63	72.33 ± 20.27	0.003 *
	Relationship with Food	18.06 ± 5.92	20.17 ± 8.39	0.290
	Food Concern	5.67 ± 1.68	7.83 ± 3.13	0.024 *
	Restriction and Compensation	6.67 ± 3.14	6.67 ± 4.21	0.416
	Feelings toward Eating	3.44 ± 1.29	3.67 ± 1.56	0.405
	Idea of Normal Eating	31.94 ± 7.41	32.75 ± 9.33	0.310

* *p*-value ≤ 0.05: statistical significance.

**Table 5 ijerph-20-01727-t005:** Mean scores differences between groups on NRS cutoff.

Scale	Subscale	No Pain/Mild Pain (M ± SD) NRS ≤ 5	Moderate/Severe Pain (M ± SD) NRS ≥ 6	*p*-Value
	Current BMI	22.24 ± 3.47	23.97 ± 8.87	0.306
BES	Tot	2.69 ± 2.024	6.71 ± 8.615	0.051
	Behavior	1.88 ± 2.06	4.14 ± 4.865	0.147
	Feelings and Cognition	0.88 ± 0.89	2.57 ± 4.274	0.027 *
EAT-26	Tot	4.25 ± 3.04	5.79 ± 3.98	0.291
	Dieting	2.38 ± 1.86	2.00 ± 2.91	0.609
	Bulimia and Food Preoccupation	0.06 ± 0.25	0.64 ± 1.646	0.010 *
	Oral Control	2.44 ± 2.16	2.43 ± 2.377	0.844
YEOQ	Tot	3.81 ± 3.92	12.64 ± 9.508	<0.001 *
	Anxiety	0.44 ± 0.73	2.14 ± 1.703	0.001 *
	Sadness	0.44 ± 0.73	1.50 ± 1.345	0.037 *
	Loneliness	0.13 ± 0.34	1.50 ± 1.286	<0.001*
	Tiredness	0.75 ± 1.13	1.29 ± 1.684	0.088
	Anger	0.31 ± 0.60	1.14 ± 1.231	0.003 *
	Happiness	0.81 ± 0.83	1.00 ± 1.038	0.222
	Boredom	0.25 ± 0.45	1.64 ± 1.082	0.003 *
	Guilt	0.25 ± 0.58	0.86 ± 1.027	0.011 *
	Physical Pain	0.44 ± 1.21	1.57 ± 1.785	0.030 *
EDE-Q	Tot	0.87 ± 0.69	2.064 ± 1.0066	0.250
DEAS	Tot	62.69 ± 10.78	75.07 ± 17.207	0.020 *
	Relationship with Food	17.06 ± 5.01	21.00 ± 8.357	0.097
	Food Concern	6.13 ± 2.68	7.00 ± 2.418	0.915
	Restriction and Compensation	6.38 ± 2.94	7.00 ± 4.206	0.159
	Feelings toward Eating	3.50 ± 1.37	3.57 ± 1.453	0.784
	Idea of Normal Eating	29.25 ± 7.87	35.71 ± 7.097	0.985

* *p*-value ≤ 0.05: statistical significance.

## Data Availability

The data presented in this study are available on request from the corresponding author. The data are not publicly available due to privacy.

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
