# Peer review of "Eating Disorders and Disturbed Eating Behaviors Underlying Body Weight Differences in Patients Affected by Endometriosis: Preliminary Results from an Italian Cross-Sectional Study"

_ijerph, 2023, doi:10.3390/ijerph20031727_

Round 1

Reviewer 1 Report

REVIEW REPORT FOR THE STUDY “EATING DISORDERS AND DISTURBED EATING BEHAVIORS UN-DERLYING BODY WEIGHT DIFFERENCES IN PATIENTS AFFECTED BY ENDOMETRIOSIS: PRELIMINARY RESULTS FROM AN ITALIAN CROSS-SECTIONAL STUDY”

Journal: Int. J. Environ. Res. Public Health

The paper "Eating disorders and disturbed eating behaviors un-derlying body weight differences in patients affected by endometriosis: preliminary results from an italian cross-sectional study", performs a study to characterise the prevalence of eating disorders (ED), binge-eating disorder (BED) and emotional attitudes to eating (EAT) in patients with endometriosis in order to understand the possible links between this influential gynaecological condition and changes in body mass index.

Title and summary. The title and abstract express well the object of study, objectives and results of the article.

Structure of the article. The contents are well organized and they adhere to the IMRaD structure. It includes a theoretical framework of the research problem but at this point I suggest the authors incorporate three bibliographic references that I miss in the text:

Rostad IS, Tyssen R, Løvseth LT. Symptoms of disturbed eating behavior risk: Gender and study factors in a cross-sectional study of two Norwegian medical schools. Eat Behav. 2021 Dec;43:101565. doi: 10.1016/j.eatbeh.2021.101565. Epub 2021 Sep 9. PMID: 34509936.

Wisting L, Skrivarhaug T, Dahl-Jørgensen K, Rø Ø. Prevalence of disturbed eating behavior and associated symptoms of anxiety and depression among adult males and females with type 1 diabetes. J Eat Disord. 2018 Sep 11;6:28. doi: 10.1186/s40337-018-0209-z. PMID: 30214804; PMCID: PMC6131775.

Elma Ö, Brain K, Dong HJ. The Importance of Nutrition as a Lifestyle Factor in Chronic Pain Management: A Narrative Review. J Clin Med. 2022 Oct 9;11(19):5950. doi: 10.3390/jcm11195950. PMID: 36233817; PMCID: PMC9571356.

Focusing the opportunity of the study, it must be said that it is a useful work since it covers the social function of informing decision-makers on public health policies about the impact of disease prevention policies (in this case eating disorders) and the promotion of health.

In that sense, authors should associate some hypotheses with the object of the study.

Materials and methods.

Regarding the material and methods section, the methodology is tailored to the object of study and the objectives and is explained in a transparent manner while it has been validly applied to guarantee the results.

However, I suggest the authors to incorporate a logistic regression that would allow the impact of the different variables on the dependent variable of the study to be assessed...

Results.

The results are significant and they are presented in an adequate and understandable way not only through narration, but also with self-explained tables and figures that are also well elaborated in terms of presentation, but it would be interesting if they incorporated source. The results justify and relate to the objectives and methods and the results are of sufficient social interest.

Discussion.

The discussion appropriately compares the study results with other works, highlighting the main study findings. 30.1% of the bibliography cited in the study belongs to the previous five years.

However, I would propose the inclusion of three bibliographic references in the discussion section:

Geller, S.; Levy, S.; Ashkeloni, S.; Roeh, B.; Sbiet, E.; Avitsur, R. Predictors of Psychological Distress inWomen with Endometriosis: The Role of Multimorbidity, Body Image, and Self-Criticism. Int. J. Environ. Res. Public Health 2021, 18, 3453. https://doi.org/10.3390/ijerph18073453

Chapuis-de-Andrade S, Moret-Tatay C, Costa DB, Abreu da Silva F, Irigaray TQ and Lara DR (2019) The Association Between Eating-Compensatory Behaviors and Affective Temperament in a Brazilian Population. Front. Psychol. 10:1924. doi: 10.3389/fpsyg.2019.01924

Santonicola A, Gagliardi M, Guarino MPL, Siniscalchi M, Ciacci C, Iovino P. Eating Disorders and Gastrointestinal Diseases. Nutrients. 2019 Dec 12;11(12):3038. doi: 10.3390/nu11123038. PMID: 31842421; PMCID: PMC6950592

The conclusions are adequately related to the objective.

Overall, it is an interesting study, and should be considered for publication in Int. J. Environ. Res. Public Health, once the minor revisions proposed have been resolved.

Reviewer 2 Report

Dear authors, 

Thanks for sending your work to this journal. Your manuscript is very interesting. But in this case, there are a few comments to improve your report. 

1. Abstract. Please add more data about the result of your work. The current form is lacking the interesting results of your work. 

2. Introduction. I know that DEBs are a good predictor/risk factor for eating disorders, this work's aim is not to measure the prevalence of ED just to know the DEBs. For that reason, I strongly suggest reducing the ED part in the introduction section. Also, I can suggest re-organizing the introduction section, first, you can write about the endometriosis diseases and symptoms, and after that, you can write about the possible relationship to DEBs. In this section you talk about endometriosis patients having lower BMI, can you add more information about it or explain why? 

3. Materials and methods. Can you please add a flow chart about the number of participants in this study? Also, explain why just 30 patients. In this section, you said: “The complete assessment of all rating scales was conducted in two different rounds” can you explain why? On different days? About statistical analysis: Please add the test name for the normality distribution analysis of continuous variables. 

4. Results. Please can you merge tables 1, 2, 3, and 4? Because all these tables are about the patient's characteristics. 

5. Discussions. Can you please extend more the ideas about pain as a mediator of eating behavior? 

Reviewer 3 Report

Dear Dr Panariello and the research team: thank you for doing this clinical research which identifies eating disorders, disturbed behaviours and emotional eating attitudes related to the weight of patients with Endometriosis. Although the overweight patient is the only one in your research, you still objectively report the result, demonstrating a statistically significant difference in physical pain score between endometriosis patients with higher BMI compared to low BMI. This is preliminary research, so the design of the research is appropriate and the statistical method is standardised and possesses certain guiding significance to the clinic. You should explore which eating guide to maintaining weight and whether the weight decrease can reduce menstrual pain in patients with Endometriosis in your next step of research. Your research should be published. 

Reviewer 4 Report

Thank you very much for the opportunity to review the study. I congratulate the authors for their very interesting results however I will make my minor comments below.

The entire work has incorrectly marked and constructed paragraphs, the paragraph should begin with an indentation in the text.

Several mechanisms have been hypothesized to explain the complex relationship between maladaptive dietary attitudes and emotional distress.” No footnote.

“EDs and the aptitude for disturbed eating such as DEBs are associated with multiple physical complications that strongly affect physical health.” No footnote.

No footnote for multiple paragraphs. Please provide citations.

The description regarding the calculation of BMI AND its classification should be in the Measures section.

Please describe in more detail the statistical methods used. Many measurements were made while the statistical description is unclear to reproduce them.

Why the measurement of height and weight was not carried out during the consultation, but introduced on the basis of the participants' declarations? This significantly affects the reliability of the results.

Table 5 and Table 6. error in the given values of p. and  M+SD

Please correct citations e.g. [62-69].

I think it would be worthwhile in the future to expand the research to include the topic of orthorexia, which is a disorder that has recently become very prevalent among women.
